# Control of Columnar Grain Microstructure in CSD LaNiO_3_ Films

**DOI:** 10.3390/molecules28041938

**Published:** 2023-02-17

**Authors:** Aleksandra V. Atanova, Dmitry S. Seregin, Olga M. Zhigalina, Dmitry N. Khmelenin, Georgy A. Orlov, Daria I. Turkina, Alexander S. Sigov, Konstantin A. Vorotilov

**Affiliations:** 1Shubnikov Institute of Crystallography of Federal Scientific Research Centre “Crystallography and Photonics”, Russian Academy of Sciences, 119333 Moscow, Russia; 2SEC “Technological Center”, MIREA—Russian Technological University (RTU MIREA), 119454 Moscow, Russia; 3Department of Materials and Technology, Bauman Moscow State Technical University, 105005 Moscow, Russia

**Keywords:** lanthanum nickelate, lead zirconate-titanate, chemical solution deposition, thin films, crystallization, transmission electron microscopy

## Abstract

Conductive LaNiO_3_ (LNO) films with an ABO_3_ perovskite structure deposited on silicon wafers are a promising material for various electronics applications. The creation of a well-defined columnar grain structure in CSD (Chemical Solution Deposition) LNO films is challenging to achieve on an amorphous substrate. Here, we report the formation of columnar grain structure in LNO films deposited on the Si-SiO_2_ substrate via layer-by-layer deposition with the control of soft-baking temperature and high temperature annealing time of each deposited layer. The columnar structure is controlled not by typical heterogeneous nucleation on the film/substrate interface, but by the crystallites’ coalescence during the successive layers’ deposition and annealing. The columnar structure of LNO film provides the low resistivity value *ρ*~700 µOhm·cm and is well suited to lead zirconate-titanate (PZT) film growth with perfect crystalline structure and ferroelectric performance. These results extend the understanding of columnar grain growth via CSD techniques and may enable the development of new materials and devices for distinct applications.

## 1. Introduction

Lanthanum nickelate LaNiO_3_ (LNO) is a conductive oxide with an ABO_3_ perovskite structure. The ability of LNO to reduce oxygen provides its wide application in catalysis, fuel cells, gas sensors, and supercapacitors [1,2,3,4]. LNO-based heterostructures have received considerable attention in the field of atomic-scale control of the heterointerfaces between different perovskite oxides to engineering new functionalities [5,6,7,8,9]. LNO is the most relevant electrode material for oxide electronics due to a relatively low electrical resistivity (100–1000 μOhm·cm) and lattice parameters close to those of many ferroelectric and multiferroic perovskites [10,11,12,13]. For this reason, ferroelectric/LNO structures are widely used to produce capacitors with improved structural and electrical performance based on ferroelectric perovskites, including lead zirconate-titanate (PZT), barium strontium titanate (BST), etc. [14,15,16]. Another important and promising property of the rare earth nickelates RNiO_3_ is their metal-to-insulator transition, which opens up possibilities for perspective applications, as it can be controlled by some external influences, including temperature, pressure, mechanical stress, and interface effects [1]. For example, Marshall et al. have demonstrated a field effect transistor with a LNO channel whose resistivity can be changed by ferroelectric PZT gate [17]. In addition, LNO is a promising material to develop passive and active electro-optical devices, including THz regions [18].

The electrical conductivity of LNO is extremely sensitive to the composition and structure of the films. Stoichiometric films are characterized by metallic behavior, while oxygen-deficient LaNiO_3-δ_ films exhibit a metal–insulator transition [19]. The dependence of conductivity on oxygen deficiency δ is attributed to the change in nickel valence from Ni^3+^ to Ni^2+^ with increasing δ [19,20,21]. The electrical properties of LNO are sensitive to mechanical stress, grain boundaries, and pores [1,22]. The resistivity of LNO films can vary from 100 to 10,000 μOhm·cm, depending on preparation techniques [23,24,25,26]. As a rule, the best properties demonstrate films with stoichiometric epitaxial or columnar grained structures.

High quality epitaxial LNO films with resistivity as low as ~100 μOhm·cm can be prepared on single-crystal substrates with similar lattice parameters (LaAlO_3_, SrTiO_3_) by molecular beam epitaxy (MBE), pulsed laser deposition (PLD), or physical vapor deposition (PVD) [27,28,29,30]. However, practical applications of such films are limited by small size substrates and high production cost. In most cases, it is required to deposit a LNO film on a structurally different substrate, including a non-crystalline one—for example, Si-SiO_2_. For this reason, chemical solution deposition (CSD) is widely used for the deposition of LNO films on substrates of various materials, size, and shape, including standard silicon wafers [15,19,31]. CSD techniques, including sol–gel, metal–organic decomposition, and other methods, are highly promising in achieving a low temperature of crystalline phase formation, improved film qualities [32,33,34], and novel structures based on a variety of materials, e.g., see refs. [35,36,37,38,39].

The most appropriate structure of LNO films grown on a substrate whose lattice is not appropriate for epitaxial growth is a columnar grain structure consisted of closely packed grains germinated throughout the film thickness. Some authors have demonstrated a columnar grain structure of LNO films using PLD [29] and CSD [15,22,40] techniques. However, columnar grain growth in LNO films is challenging. Most of the authors who studied the CSD of LNO films on substrates of various structures reported films with a porous, polycrystalline structure and fine equiaxed grains [19,24,41,42]. The defective structure causes enhanced resistivity, and it is not optimal for growing on the ferroelectric perovskite layer or other functional heterostructures [22].

In this work, we carry out a detailed structural analysis of LNO films prepared on a Si-SiO_2_ substrate at various stages of crystal structure formation, visualizing for the first time the mechanism of columnar grains growth on a non-crystalline surface. Together with the structural analysis, the analysis of the local elemental composition allows us to give a new insight on the crystallization kinetics of LNO films prepared by CSD techniques.

Columnar grain growth in CSD films is usually caused by heterogeneous nucleation at the film–substrate interface [34,35,36,37,38,39,40,41,42,43,44,45]. For this reason, the main strategy of columnar grain growth in CSD films is a choice of the substrate with a close lattice constant or introducing an intermediate seeding layer whose structure decreases the Gibbs free energy Δ*Gv* at the interface [32,46,47]. Here, we demonstrate another mechanism: the coalescence of small grains by successive layer-by-layer deposition. In contrast to heterogeneous nucleation, this method is successfully realized on a non-crystalline SiO_2_ surface, which opens wide applications for it in various structures and devices. One such ferroelectric LNO/PZT structure prepared on the Si-SiO_2_ substrate with good electrical performance is demonstrated in this work.

## 2. Results and Discussion

### 2.1. Effect of Annealing Time on the LNO Microstructure

To study crystallization kinetics, we have prepared LNO films with different annealing time of each layer (2, 5, 7, 10, or 20 min). The same processing time was used during the formation of all seven layers upon layer-by-layer deposition (see Section 3. Materials and Methods).

Figure 1 shows the microstructure of the film prepared by annealing of each LNO layer during 2 min (LNO-cryst-2 min). The film is non-uniform with a pronounced surface relief; the thickness varies from 100 to 180 nm (Figure 1a). The film consists of alternate crystalline parts with minimum thickness and amorphous regions with maximum thickness (Figure 1b). The electron diffraction patterns from the region with and without the hillock are shown in Figure 1c,d, where the diffraction pattern from the amorphous region contains a characteristic halo, which is absent for the case of a flat polycrystalline region. The presence of diffraction reflexes from an amorphous region can be associated with both the presence of some crystals inside it and the capture of neighboring regions in the field of a selective aperture.

The crystalline region consists of non-oriented crystallites of the cubic LaNiO_3_ phase (s.g. $Pm\bar{3}m$) with an average grain size of 6–22 nm. However, there are both large grains up to 45 nm and fine grains of 1–3 nm in size. Similar equiaxed grains have been observed earlier in our work [48], as well as by other authors, with the use of different processing conditions [13,19,49]. Crystallites are located mainly within the deposited individual layers, between which there are pores.

According to the EDX analysis, the composition of amorphous hillocks does not differ from the composition of the crystalline areas (Figure 2). However, the elemental maps show a slight fluctuation of the lanthanum and nickel content, perpendicular to the substrate direction. The detected fluctuations in the chemical composition correlate with the layer-by-layer deposition of precursor solutions. A region at the top part of each layer is slightly enriched with La, whereas at the bottom part it is enriched with Ni. These fluctuations are likely formed due to NiO evaporation and an interaction of the film surface with the environment during the heat treatment [50,51,52,53,54,55].

To check the reason for hillocks relief formation, we have studied surface of samples just after spin-on deposition and after baking at different temperatures. Fresh films and films after baking at 200 °C, 10 min demonstrate a flat surface, see Figure 3a. A relief of hillocks appears as a result of heat treatment at 300 °C, 10 min (Figure 3b). Thus, this interesting phenomenon is caused not by the spin-on deposition of liquid precursor, but is unambiguously initiated by some transformations during film drying.

The microstructure of a cross section of the multilayer film after drying at 300 °C is shown in Figure 4. The HAADF STEM image (Figure 4a) shows a difference in contrast of peaks and valleys. The contrast in HAADF STEM images depends on the z-number of elements and the material density. According to the elemental maps (Figure 4b,c) and SAED patterns from the valley (Figure 4d) and the peak (Figure 4e), the elemental composition is the same in both peaks and valleys. Thus, we can conclude that the contrast of the film is determined only by the density. The film structure is denser in the hillocks, and less dense with many pores in the cavities. The above-described La/Ni composition inhomogeneity are clearly manifested as well (Figure 4d,e).

The sheet resistance of the films prepared by the layer-by-layer deposition and annealed for 2 min appeared to be very high *R_s_* = 1078–2862 Ohm/square due to uncompleted crystallization of the film. Taking into account the film thickness, it corresponds to the resistivity *ρ* = 18,000–30,000 μOhm·cm.

An increase in the annealing time from 2 to 5 min leads to the formation of columnar crystallites with the width of 25–63 nm (Figure 5). Columnar grains grow mainly from the substrate. There is approximately an equal number of grains grown completely and partially over the entire film thickness. However, layers with fine-grained equiaxed grains of 7–19 nm in size (Figure 5a,b) are still present throughout the film volume. Pores are also located in these areas. An estimated film thickness is 100–160 nm. According to the high-resolution images (Figure 5c,d), the film is crystallized mainly with the formation of the cubic perovskite LaNiO_3_ phase (s.g. $Pm\bar{3}m$), although it should be noted that part of the Fourier diffractograms can be solved using both the cubic and trigonal LaNiO_3_ ($R\bar{3}c$). A striation is traced parallel to the substrate plane in STEM HAADF images (Figure 5b) characterized by z-contrast and density contrast.

A surface relief in the LNO-cryst-5 min sample retains. However, hillock areas contain small grains and pores instead of the amorphous region in the LNO-cryst-2 min sample. As a result of complete crystallization, the sheet resistance decreases drastically to *R_s_* = 86 ± 6 Ohm/square (the resistivity *ρ* = 970–1130 μOhm·cm).

With the further increase in the annealing time to 7 min, the grain structure becomes predominantly columnar with the presence of a small amount of equiaxed crystallites (Figure 6a,b). The average width of columnar grains is 28–48 nm, while equiaxed grains are 8–18 nm in size. The film thickness is 90–120 nm. A decrease in the film thickness while maintaining the same number of layers may indicate a compaction of the structure. The lower area of the film is more porous (Figure 6c). The HAADF STEM images show striations placed parallel to the substrate plane (Figure 6c). Such a variation in contrast may be caused by a change in density or composition (z-contrast).

According to the SAED patterns and HRTEM images (Figure 7), the main phase is cubic perovskite LaNiO_3_ $Pm\bar{3}m$, however, SAED patterns contain reflections that do not belong to the main phase. These reflections (corresponding to interplanar distances *d_hkl_* = 3.44, 3.15, 2.44, 1.54 Å) belong to neither platinum (protective layer) nor gallium (from ion etching of the FIB sample). The reflections corresponding to *d_hkl_* = 2.44, 1.54 Å may refer to the nickel oxide NiO (s.g. $Fm\bar{3}m$), whereas *d_hkl_ =* 3.44, 3.15 Å may indicate the presence of the lanthanum oxide phase La_2_O_3_ (s.g. $P\bar{3}m1$) or other phases. However, these phases were not detected by X-ray phase analysis or localized by elemental analysis (EDX) and HRTEM. Thus, it can be assumed that the number of these phases is very small.

The sheet resistance of the films after annealing for 7 min does not practically change: *Rs* = 90.5 ± 4.2 Ohm/square, or in terms of resistivity, *ρ* = 900–990 µOhm·cm.

Increase in the layer annealing time up to 10 min leads to the formation of columnar grains of 22–38 nm in size in all areas (Figure 8). The film has a strongly pronounced relief; its thickness varies from 65–90 nm in relatively flat areas to about 160 nm at the hillock’s tops. Pores and their agglomerates are located at the hillock’s boundaries and near the film interface. In the SAED pattern (Figure 8b), all strong reflexes belong to the LaNiO_3_ (s.g. $Pm\bar{3}m$) phase, however, there are a couple of very weak reflexes similar to the previous film: 3.44 and 3.17 Å. In addition, a pair of reflexes is observed, corresponding to 2.83 Å. Interplanar distances of 3.17 and 2.83 Å can correspond to the tetragonal La_2_NiO_4_ phase (s.g. I4/mmm). HRTEM images indicate the predominant presence of the cubic perovskite phase (Figure 8c,d).

The sheet resistance of the film after annealing for 10 min is *R_s_* = 88.0 ± 3.3 Ohm/square, which corresponds to the resistivity *ρ* = 680–730 μOhm·cm.

Pores accumulation occurs mainly at the bottom part of the film in the sample crystallized for 20 min, as it is shown in Figure 9. This porous region destroys uniformity of the grain structure: small grains are located at the film surface, whereas the columnar grains grow higher (Figure 9a,b). The film thickness ranges from 70 nm at the thinnest points to 160 nm at the highest ones. The width of the columnar grains is 21–43 nm. Striation in STEM images is also preserved (Figure 9a). HRTEM images correspond to the cubic perovskite phase LaNiO_3_ (Figure 9c,d). In general, the structure of samples annealed for 10 and 20 min differs slightly, except of the pores’ localization.

The sheet resistance of the film after annealing for 20 min is 85.6 ± 5.7 Ohm/square, which in terms of resistivity is *ρ* = 920–1100 μOhm·cm.

Thus, structural studies of the crystallization kinetics showed that LNO tends to form equiaxed grains at the brief annealing of each spin-on deposited layer for 2–5 min. Longer annealing time results in the columnar grain structure. Agglomeration of pores inside LNO film can locally destroy the columnar structure. A slight fluctuation of La and Ni content in the perpendicular to the interface direction is observed and suggests some interaction of the deposited layer with the environment during film formation. A sharp decrease in the film resistivity is detected at transition from the amorphous state to the crystalline one. However, the change in the crystal structure from fine-grained to columnar is not accompanied by a noticeable change in the resistivity of the film.

### 2.2. Structure of Samples with Additional Annealing

LNO film was deposited on Si-SiO_2_ (10 nm) substrate by seven deposition steps with annealing of each layer at *T* = 650 °C for 10 min. After deposition, the sample was divided into three parts, each of which was additionally annealed at *T* = 650 °C for 30, 60, and 90 min, respectively.

All prepared films demonstrate a specific relief of the film surface, described above, which is noticeable both in the SEM images (Figure 10a) and the TEM cross-section image (Figure 10d). In addition, the SEM images clearly show a network of crack-like defects (Figure 10a). The authors of ref. [56] attribute the appearance of cracks to stresses caused by intense decomposition and removal of organic components during rapid heating of the films. However, in the BF STEM image of the plan-view sample (Figure 10b), one can see a chain of pores instead of cracks. Apparently, the formation of pores during crystallization and their subsequent propagation can be the reason for the formation of crack-like defects. All obtained samples of this series have a thickness of about 120 nm and are crystallized with the formation of columnar grains of 18–72 nm in size (Figure 10c).

In the BF TEM image (Figure 10d), a specific contrast is observed inside individual columnar grains, which can be associated with small misorientations of the crystal lattice (blocks) inside the grain. In contrast to the samples with crystallization of each layer for 10 and 20 min from the previous series, the LNO films in this series have practically no pores in the bulk and have a perfect columnar structure. Thus, an additional annealing of the films leads to improvement of their crystalline performance.

For an informative electron diffraction pattern, a plan-view lamella of an LNO-anneal-90 min sample was prepared (Figure 10b,c). According to the obtained electron diffraction patterns, the sample crystallizes with the formation of a cubic phase (s.g. $Pm\bar{3}m$), however, several electron diffraction patterns showed single reflections corresponding to interplanar distances of 3.27 Å, 2.43 Å, 1.68 Å, and 1.46 Å. Interplanar distances of 2.43 Å and 1.46 Å may indicate the presence of a residual amount of NiO (s.g. $Fm\bar{3}m$, ICSD ID 9866, *d* = 2.43 and 1.49 Å), while 3.27 Å and 1.68 Å correspond to La_2_O_3_ (s.g. $Ia\bar{3}$, PDF 00-022-0369, *d*= 3.27, 1.70, 1.66 Å), often observed in LNO films obtained by CSD [15,57,58].

The La:Ni ratio according to elemental mapping is 51.2:48.8 (integrated over the region with dimensions of 120 × 370 nm). However, fluctuations of the lanthanum and nickel content are still observed at the elemental map (Figure 10b) and results of the quantitative analysis of the elemental composition along the line. In this case, the fluctuations are not so large; according to the data of EDX analysis along the line, the change in the amount of both La and Ni is within 5 at.%.

Formation of the columnar structure on the amorphous SiO_2_ layer is not trivial and thus requires the detailed structure research. In all films, an amorphous sublayer of approximately 5 nm thick is observed at the SiO_2_/LNO interface. The similar region was reported in [15], however without thorough structure and chemical analysis. The STEM images clearly show a chain of pores at the film/substrate interface of 4–12 nm in size, as well as single pores in the film bulk (Figure 11a). In addition, the 2–3 nm LNO layer adjacent to SiO_2_ is saturated with lanthanum, and above it there is a zone enriched in nickel with a thickness of about 7 nm (Figure 11b). The local elemental mapping presented in our work shows that there is no diffusion of silicon into the LNO layer. Regardless, additional annealing for 30–90 min does not completely eliminate compositional inhomogeneity, however, it gives well-packed columnar grains.

The sheet resistance of LNO films demonstrates some increase with the increase in annealing time duration: *R_s_* = 86.0 ± 5.0, 99.5 ± 2.3, 103.0 ± 5.9 Ohm/square at annealing 30, 60, and 90 min, respectively.

Table 1 summarizes the data of the performed structural analysis.

Figure 12 shows the film resistivity as a function of annealing time of each layer. The most dramatic ten-times decrease in the film resistivity occurs at the change in annealing time from 2 min to 5 min. Obviously, this is due to the disappearing of amorphous regions and complete crystallization of the film volume. Resistivity of completely crystalline films prepared at higher annealing time differs slightly. However, the increase in annealing time from 5 to 10 min slightly decreases the resistivity value, and this can be caused by improving of the film crystalline performance [1,22]. The columnar structure of the film prepared at the 10 min annealing contains less internal boundaries, and this can affect the free path of charge carriers. The increase in annealing time of each layer up to 20 min or additional annealing of the film at 30–90 min led to a slight increase in resistivity, see Figure 12. As it is known, oxygen deficiency is the reason for the resistivity increase attributed to the change in nickel valence [19,20,21,59]. Another possible reason could be the increase in mechanical stress, causing charge carriers transport. This suggestion needs more detailed study, including XPS and temperature dependence of resistivity data, which is out of scope of this paper. Nevertheless, the film prepared at 10 min annealing demonstrates the lowest resistivity.

### 2.3. Structure of PZT-LNO-SiO_2_-Si Compositions

The PZT film was deposited on the LNO layer obtained with additional annealing for 30 min. The structure of PZT-LNO-SiO_2_-Si composition is shown in Figure 13. Both layers are quite dense and demonstrate a columnar structure. Figure 13b shows a DF TEM image of a pair of mutually oriented LNO–PZT grains. The electron diffraction taken from this region and the HRTEM image of the PZT/LNO interface (Figure 13c,d) show that the (100) and (011) planes in PZT and LNO are oriented in the same way. Thus, the LNO layer perfectly matches PZT perovskite crystallites serving as a seed layer.

Electrical properties of ferroelectric PZT capacitors prepared on the Si-SiO_2_-LNO substrate were studied in comparison to the similar PZT film deposited on the standard Si-SiO_2_-Ti-Pt substrate. Figure 14 shows the dependences of the polarization, switching current, and dielectric constant on the electric field strength. Some parameters extracted from Figure 14 are collected in Table 2. Here, *P*_r_^(+)^, *P*_r_^(−)^, *E*^(+)^, *E^(^*^−)^ are the remanent polarization and the coercive field at positive and negative directions, respectively, extracted from dielectric hysteresis loops *P*(*E*). Further, *ε _max_^(+)^, ε _max_^(^*^−*)*^*, E^(+)^, E^(^*^−)^ are the maximum values of permittivity and corresponding value of electric field at positive and negative directions, respectively, extracted from capacitance–voltage dependences recalculated to the permittivity–electric field dependence *ε (E*).

Generally, PZT-LNO and PZT-Pt capacitors have similar electrical properties, except for asymmetry and a somewhat higher value of coercive field in the case of Pt electrode. The reason for this difference lies, first of all, in the difference of interface properties. Ferroelectric/metal interfaces can be both impenetrable and transparent for oxygen vacancies [60]. PZT-Pt interfaces blocks oxygen vacancies that leads to their accumulation near the interface, whereas PZT-LNO and PZT-Hg interfaces are transparent for them. Defect chemistry is most responsible for the electrical properties of PZT films, including leakage currents, fatigue, retention, imprint, etc. [61]. In the case of impenetrable interfaces, the accumulation of oxygen vacancies can stimulate electrons injection with the formation of induced *p-n* junctions [62]. Another possible reason is the different dead layer thickness at the interfaces [63].

### 2.4. Mechanism of Crystallization

Crystalline structure transformation with the increase in annealing time of each layer at the layer-by-layer spin-on deposition of LNO film is shown in Figure 15. LNO film prepared at 2-min annealing consists of alternating amorphous and polycrystalline areas, corresponding to peaks and valleys in the film relief (Figure 15a). Crystalline regions contain fine randomly orientated crystals and many pores between them. Film crystallization completes after 5 min annealing with the formation of column grains and presence of small equiaxed grains at the peak tops (Figure 15b). The increase in the annealing time leads to an entirely columnar grain structure. The crystalline performance was improved with the annealing time increase (Figure 15c,d). However, annealing at 20 min leads to the appearance of pores agglomerates in the bottom part of the film, disturbing column grain uniformity (Figure 9). It seems that the reason for this may be defects gettering caused by mechanical stress and temperature impact [64]. Long post annealing of LNO films improves crystalline performance, however, chains of pores are still observed inside the film.

Based on the obtained structural data, we can discuss transformation of the film structure at LNO spin-on deposition. The main issues of this discussion are as follows:

Why is columnar grain crystalline structure formed in LNO film deposited on the amorphous substrate without any seed?

What is the reason for the relief formation?

A simplified model of this process is shown in Figure 16. The following processing steps could be distinguished and discussed below.

Spin-on deposition of precursor’s solution. Solutions of La(CH_3_CO_2_)_3_·xH_2_O and (Ni(OCOCH_3_)_2_·4H_2_O dissolved in acetic acid CH_3_COOH are deposited on a silicon wafer. The liquid moves on the spinning wafer under an action of centrifugal force and viscous resisting force [65,66]. Film thinning causes a mass transfer increase via CH_3_COOH evaporation, which promotes interaction between acetate species. As a result of viscosity increase, liquid outflow stops, and further film thinning occurs only due to liquid evaporation. Acetic acid and water released from acetate molecules are partially evaporated (boiling point of acetic acid T = 118 °C). The film demonstrates a flat surface without any relief after spin-on deposition and after soft bake at 200 °C. Metal acetates LaAc_3_ and NiAc_2_ completely or partially lose water during the soft bake. Based on TGA data, Hussein shows two peaks at 130 °C and 180 °C due to the following LaAc_3_ dehydration reaction [67]:


(1)
$$\mathrm{La}{\left({\mathrm{CH}}_{3}\mathrm{COO}\right)}_{3}\cdot 1.5H_{2}O\to \mathrm{La}{\left({\mathrm{CH}}_{3}\mathrm{COO}\right)}_{3}+1.5H_{2}O$$


Partial dehydration of NiAc_2_ gives Ni hydroxides and acetates (1−x) Ni(CH_3_COO)_2_·×Ni(OH)_2_ as an intermediate product [68,69,70]. It should be noted that the soft bake at 200 °C does not lead to the creation of metal oxides that could interact and create new phases. According to XPS data of Jesus et al. [70], the Ni carboxylate component diminishes strongly after heating at 300–350 °C and disappears on heating at 450 °C. If drying proceeds in this temperature range, an interaction between metal oxide molecules occurs and crystallization creates not only LaNiO_3_ (s.g. $Pm\bar{3}m$), but also Ni_3_O_4_ (*C2/m*), La_2_O_3_ (*C2/m*) and La_2_Ni_2_O_5_ (*C2/m*) phases [71]. Thus, soft-baking at a low temperature ~ 200 °C is a key processing step, as hard-baking at higher temperature gives the creation of various undesirable phases.

2.Heat treatment at 300 °C. Note, that this is not a film processing step. However, this intermediate state is important to clarify why the films demonstrate wavy shape surface relief. As it was discussed earlier, an increase in temperature up to 300 °C leads to the formation of a pronounced relief with a regular wave structure (see Figure 3). After dehydration, LaAc_3_ have a tendency to association. For example, Hussein points to LaAc_3_ recrystallization at 210 °C [67]. NiAc_2_ dissociation starts in the same temperature range [70]. Regardless, we can consider the association of acetate and hydroxide precursors with the formation of small particles or clusters. We propose that the relief is formed as a result of self-assembly in the “solvent–particles” system caused by the evaporation of the solvent and interaction between particles (e.g., van der Waals interaction) [72,73]. The phenomenon of self-assembly in the dispersion of solid particles was first noticed during the drying of coffee drops, when the particles migrate to the edge of the drop and form a dark ring. Since then, this phenomenon has been shown for a wide range of surfaces, solvents, and solutes, and the organization mechanism has been shown to be caused by outward flow, which is driven by solvent loss through evaporation and geometric constraints [74,75]. It was shown that systems far from equilibrium can exhibit complex transitory structures, even under equilibrium fluctuations [72,73,76,77,78,79]. We suggest that a similar self-assembly process is the reason for the regular wavy shape surface in LNO films. More detailed investigation of this phenomenon may be useful in view of the creation of structures with special regular form.3.High temperature annealing at 650 °C. The increase in temperature results in the transformation of LaAc_3_ and NiAc_2_ into La and Ni oxides. The sequence of this transformations may be as follows [67,70,80,81]:


(2)
$$2\mathrm{La}{\left({\mathrm{CH}}_{3}\mathrm{COO}\right)}_{3}\to {\mathrm{La}}_{2}O_{2}{\mathrm{CO}}_{3}+3{\mathrm{CH}}_{3}{\mathrm{COCH}}_{3}+2{\mathrm{CO}}_{2}$$



(3)
$${\mathrm{La}}_{2}O_{2}{\mathrm{CO}}_{3}\to {\mathrm{La}}_{2}O_{3}+{\mathrm{CO}}_{2}$$



(4)
$$\mathrm{Ni}{\left({\mathrm{CH}}_{3}\mathrm{COO}\right)}_{2}={\mathrm{NiCO}}_{3}+{\mathrm{CH}}_{3}{\mathrm{COCH}}_{3}$$



(5)
$${\mathrm{NiCO}}_{3}=\mathrm{NiO}+{\mathrm{CO}}_{2}$$


LNO perovskite phase can be created due to oxide species interaction:(6)$${\mathrm{La}}_{2}O_{3}+4\mathrm{NiO}+{\;O}_{2}\to 4{\mathrm{LaNiO}}_{3}$$

It should be pointed out that at 650 °C, annealing the Gibbs free energy Δ*Gv* is the minimum for pure LaNiO_3_ (s.g. $Pm\bar{3}m$) perovskite phase creation (see Figure 2). In contrast, in the case of hard-baking at 300–450 °C, intermediate phases are created and final annealing at 650 °C gives the multiphase structure, as it was mentioned before [71].

Crystallization starts in the valley parts of the film, where film density is lower and oxide species have higher mobility for interaction and crystal growth (see Figure 15a). With increasing annealing time, equiaxed grains are created throughout the film volume. As it was pointed earlier (see Figure 2), the film surface changes in composition due to NiO evaporation and interaction with the atmosphere [50,51,52,53,54,55].

4.Deposition of the next layer. The upper layer is deposited under the same conditions as the previous one. An increase in annealing time leads to the coarsening and formation of column grains via crystallites coalescence. From a thermodynamic point of view, the driving force of coalescence is the reduction in the surface free energy. According to the pioneered Frenkel theory, the coalescence kinetics of two identical touching droplets is governed by viscous flowing and depends on their surface energy, viscosity, and size [82]. The characteristic time of coalescence 𝜏 followed the linear dependence on the initial particle radius 𝑟_0_: *τ = η σ 𝑟_0_*, where 𝜂 is the shear viscosity coefficient, and 𝜎 is the surface tension [82,83,84]. In our case, the processing temperature (650 °C) is lower than the melting point (~2200 °C), and the coalescence mechanism relates rather to the surface or grain boundary diffusion than to the hydrodynamic flows inside the droplet; however, Frenkel’s approach is still valid for rough process characterization [83]. Thus, sintering via coalescence of individual crystals is a time-dependent process and, in the case of LNO film, requires more than 5 min annealing of each layer (see Figure 16). As each deposited layer undergoes baking at 200 °C and annealing at 650 °C, the total annealing time of bottom layers will be significantly higher (35 min for the first deposited layer). As coalescence time τ is proportional to the crystallites size 𝑟_0_, the rate of coalescence is reduced with the crystallites growth and the coalescence of bigger crystals requires more time (10 min or more for each layer to complete column grain formation). It should be noted that sintering depends on the crystalline orientation of particles as the grain boundary energy relates to the degree of misorientation between two particles and gives an energy barrier for particle coalescence [83,85]. Reorientation takes some energy and time; for this reason, we have observed some disorientation inside the column grains as it was mentioned earlier.

Usually, creation of the columnar grain structure in thin films is governed by hete ogeneous nucleation at the substrate or seed layer, e.g., see [46]. The described self-seeding process via coalescence of small grains through the layer-by-layer deposition and annealing is unique and permits to grow films with column grains on any substrates.

### 2.5. Electrical Properties

As it is expected, the columnar grain LNO structure is ideally suited for PZT growth. The LNO/PZT interface demonstrates the epitaxial growth of PZT grains without any transition zone (Figure 13d). The columnar structure of PZT film on the columnar grain LNO layer is much more uniform than on the LNO with the equiaxed grain structure [48]. In the last case, a large amount of small and randomly oriented LNO crystallites causes the nucleation of many PZT crystallites that are competing with each other. As a result, many grains do not grow through the entire film thickness [48]. Thus, the columnar structure of the LNO layer is preferable to obtain better crystalline performance of PZT film.

Another important and non-obvious issue is the effect of the LNO structure on the resistivity of the film. It is known that LNO in stoichiometric composition demonstrates metallic conductivity (the dominant charge carriers contributing to transport in LaNiO_3_ are holes), although Ni has a formal valence of 3+ [59]. Oxygen deficiency in LaNiO_3-δ_ leads to the creation of Ni^2+^ ions, significantly affecting the conductivity [59,86]. For instance, resistivity can change by an order of magnitude or more when δ deviates from zero by as little as 0.1 [86]. With a decrease in the Ni^3+^/Ni^2+^ ratio, the LNO demonstrates semiconducting behavior as a result of band gap appearance, e.g., see [87]. In this study, we do not perform estimation of the Ni^3+^/Ni^2+^ ratio, as here we focus mainly on the films’ crystalline structure. From this point of view, we can analyze data on LNO films resistivity.

Figure 17 presents resistivity of LNO films data collected from literature. The data are divided in accordance with their microstructure: bulk, and films with a single crystal, columnar, and equiaxed polycrystalline structure. It is hard to perform rigorous treatment of these data, as various deposition methods, annealing conditions, and measurement techniques are used by different researchers. However, we can conclude that LNO resistivity increases in the following line: bulk < epitaxial < column < equiaxed. The reported values of resistivity at room temperature are *ρ* ≈ 90 µOhm·cm for single crystal bulk LNO [88] and *ρ* = 380 µOhm·cm for powder bulk samples [89]. These are rather low *ρ* values, which is attractive for many applications. Of course, this value is too high for BEOL (back-end-of-line) in semiconductor manufacturing, but it is comparable with the resistivity value of some metals (e.g., *ρ* = 47.8 µOhm·cm for titanium) or metal silicides, and thus LNO films can be used as contact and barrier layers. Epitaxial thin films obtain approximately the same resistivity from *ρ* =100 µOhm·cm for the PLD technique [28] to *ρ* = 340 µOhm·cm for the CSD method [90]. However, epitaxy requires substrate with similar lattice parameters. Unfortunately, if silicon wafer is used as a substrate*,* the *ρ* value of LNO increases: the minimum obtained *ρ* values increase by 3–4 times—*ρ* = 420 µOhm·cm—for PLD [27], and *ρ*~700 µOhm·cm for the CSD method [this work]. Films with equiaxed grains have somewhat higher resistivity values *ρ* = 900–1050 µOhm·cm for the CSD technique [19,42], this work. As the PLD technique is not appropriate for deposition on standard silicon wafer, it seems that the CSD technique of column grain LNO preparation is a good choice for different applications in electronics.

It should be noted again that we do not here discuss transport mechanisms in these films. The film resistivity depends strongly both on oxygen deficiency and scattering of charge carries on the grain boundaries (the mean free path of charge carries in LNO is about 30–40 Å [91]). It requires some special experiments and characterization techniques (see e.g., [22,51,59,86,87,91]), and this is beyond the scope of this paper.

## 3. Materials and Methods

LNO film-forming solution was prepared according to the method developed by N. Kotova [48]. The solution of lanthanum (III) acetate hydrate (La(CH_3_CO_2_)_3_·xH_2_O, 99.9%, Sigma-Aldrich, Steinheim, Germany) and nickel (II) acetate tetrahydrate (Ni(OCOCH_3_)_2_ ·4H_2_O, 99.998%, Sigma-Aldrich, Steinheim, Germany) in glacial acetic acid (CH_3_COOH, 99.99+%, Sigma-Aldrich, Steinheim, Germany) had the molar concentration of precursors 0.2 mol/L; the molar ratio of La:Ni was 1:1.

PZT solution was prepared by the technique described earlier [93]. Zirconium (IV) isopropoxide isopropanol complex (Zr(OCH(CH_3_)_2_)_4_·(CH_3_)_2_CHOH, 99.9%, Sigma-Aldrich, Steinheim, Germany), titanium (IV) isopropoxide (Ti(OCH(CH_3_)_2_)_4_, 99.999%, Sigma-Aldrich, Steinheim, Germany) and anhydrous lead acetate (Pb(CH_3_COO)_2_) obtained by PbO reacting with acetic acid in acetic anhydride [93] were used as precursors. The synthesis was performed in 2-methoxyethanol (CH_3_OCH_2_CH_2_OH, ≥99.9%, Sigma-Aldrich, Steinheim, Germany). The final concentration of the film-forming solution in terms of the sum of titanium and zirconium alkoxides was 0.25 M. To compensate the PbO loss during the film annealing, 15 wt.% Pb excess was added to the precursor solution. The Zr/Ti ratio was chosen as 52/48, near the morphotropic phase boundary, where PZT demonstrates enhanced piezoelectric and ferroelectric performance [94].

LNO and PZT solutions were spin-coated using a Spin-150i NPP (Semiconductor Production Systems, Ingolstadt, Germany). Silicon wafers (B-doped, (100), 12 Ohm·cm) with 10 nm or 500 nm SiO_2_ layer were used as substrates. To obtain the required film thickness, we have used a successive deposition of several layers with annealing of each layer (layer-by-layer method). This technique permits one to increase film thickness without its cracking, as a result of stress release generated in films at its heat treatment [95]. Seven layer-by-layer depositions were used to form LNO films. Each LNO layer was soft-baked at 200°C for 4 min and annealed at 650 °C from 2 to 20 min. Some samples were subjected to additional annealing at 650 °C for 30, 60, and 90 min after formation of the last layer.

PZT film consisted of 10 layers with the soft-baking of each layer at 200 °C, 4 min and hard-baking at 400 °C, 10 min. Final annealing to perform film crystallization was carried out after applying every fifth layer at 650 °C, 10 min to prevent cracking. The details of sample preparation are presented in Table 3.

Film microstructure was studied by transmission electron microscopy (TEM) in bright-field (BF) and dark-field (DF) modes, selected-area electron diffraction (SAED), high-resolution TEM mode, high-angle annular-DF scanning transmission electron microscopy (HAADF-STEM), and energy-dispersive X-ray analysis using an electron microscope (Tecnai Osiris, Thermo Fisher Scientific, Waltham, MA, USA) operated at an accelerating voltage of 200 kV.

Cross-sections of specimens and plan-view samples for TEM were prepared using a focused ion beam in a scanning electron microscope (Scios DualBeam, Thermo Fisher Scientific, Waltham, MA, USA). For this purpose, a protective Pt coating was applied to a selected area of the film surface in the modes of an electron beam (e-Pt) and an ion beam (ion-Pt) using a gas injection system. A lamella with dimensions of 15 × 1.5 × 7 μm was cut from the sample perpendicular to the surface. The lamella was then transferred using a built-in manipulator and attached to a four-post Cu lift-out grid by Pt deposition. The lamella was subsequently thinned with a focused ion beam to a thickness of 50 nm; the voltage of the ion beam was initially 30 kV and was gradually lowered to 5 kV as the specimen thickness decreased. To prepare a plan-view lamella, i.e., parallel to the substrate, the technique from Ref. [96] was used.

The sheet resistance of the films was measured using a Cresbox semi-automatic four-point probe sheet resistance measurement system (Napson, Tokio, Japan) using the standard four-point probe direct current method. The obtained sheet resistance value (*R_s_*, Ohm/square) can be used to calculate resistivity (*ρ*, μOhm·cm) as $\rho =R_{s}\cdot d$, where *d* is the film thickness [97].

Dependences of the spontaneous polarization *P*, switching current *I*, and permittivity ɛ on the electric field strength *E* for ferroelectric structures were obtained using Agilent 4284A (Santa Clara, CA, USA) and AixACCT TF Analyzer 2000 (Aachen, Germany). An MDC 802B-150 (Chatsworth, CA, USA) mercury probe was used to create a top contact to the PZT film with a dot area of 0.515 mm^2^.

## 4. Conclusions

The given research focuses on mechanisms of crystalline structure formation and their control in LNO films prepared by chemical solution deposition on a standard Si wafer covered by a non-crystalline SiO_2_ layer. The LNO layer was obtained via spin-on deposition of acetate precursors solution with soft-baking at 200 °C, 4 min and heat treatment at 650 °C during different times from 2 to 20 min. The deposition and annealing sequence was repeated until the film reached the required thickness. The annealing time of each deposited layer is the key factor governing the film crystalline structure. According to TEM analysis, annealing of each layer at less than 5 min produces small equiaxed grains, whereas longer annealing time leads to a columnar grain formation. The soft-baking temperature should be kept as low as 200 °C to produce pure perovskite LaNiO_3_ (s.g. $Pm\bar{3}m$) phase, while in the case of hard-baking at 300–450 °C, intermediate phases are created, and final annealing at 650 °C gives undesirable phases. The columnar grain microstructure is obtained as a result of small grain coalescence through the successive layer-by-layer deposition and annealing of each deposited layer.

LNO film with a columnar structure is an ideal bottom electrode for ferroelectric PZT capacitors with excellent electrical performance, as it provides epitaxial growth of column perovskite grains in ferroelectric film without any disturbing layer at the LNO–PZT interface. Columnar grain LNO films demonstrate electrical resistivity as low as *ρ*~700 μOhm·cm, whose value is attractive for various applications in silicon electronics.

In addition, we have observed a wave-shaped surface relief formation, which takes place as a result of solvent evaporation at T~300 °C and self-assembly interaction of acetate species. We suggest that further investigation of this phenomenon may offer a way to create structures with controllable shape.

## Figures and Tables

**Figure 1 molecules-28-01938-f001:**
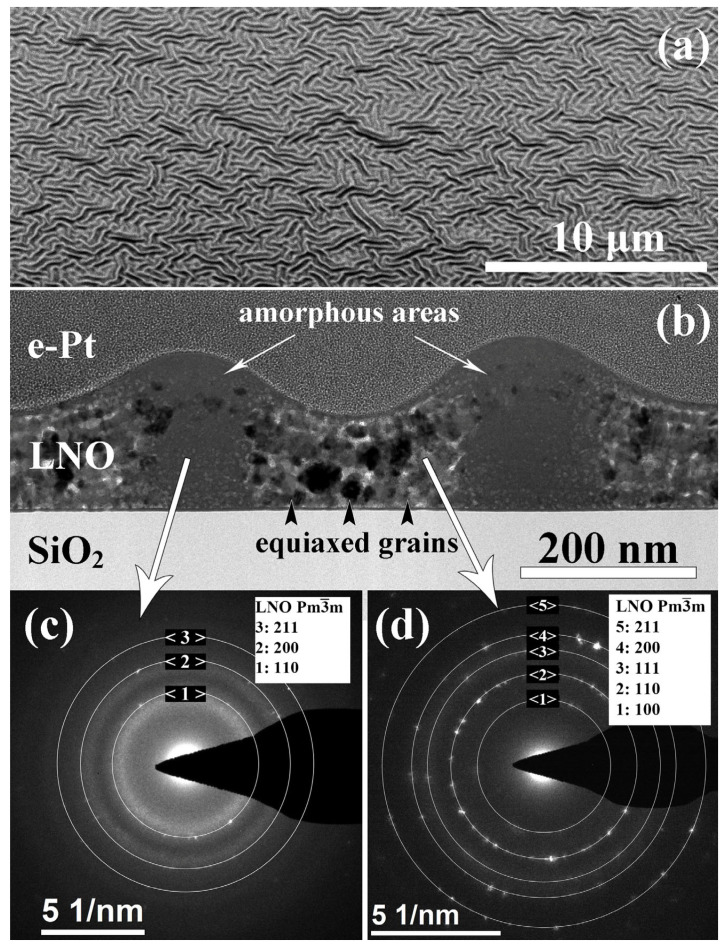
Microstructure of the cross section of LNO-cryst-2min sample: SEM image of the surface at the angle of 52° (**a**); bright-field TEM image (**b**), electron diffraction patterns from the amorphous (**c**) and crystalline (**d**) regions.

**Figure 2 molecules-28-01938-f002:**
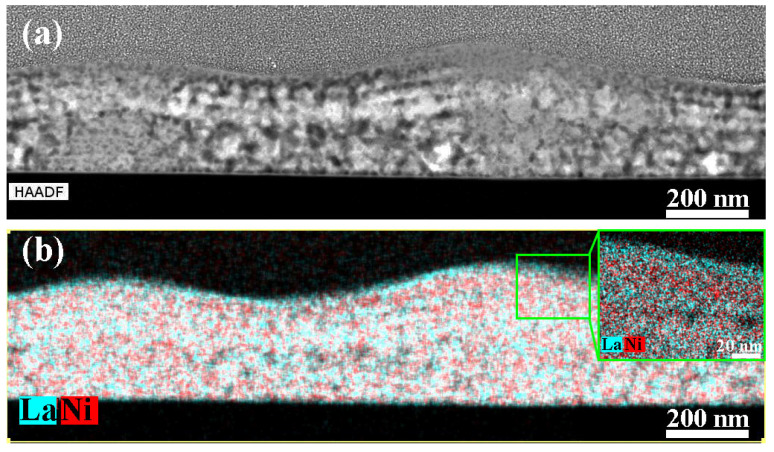
HAADF STEM image (**a**) and elemental maps (**b**) of LNO-cryst-2 min sample.

**Figure 3 molecules-28-01938-f003:**
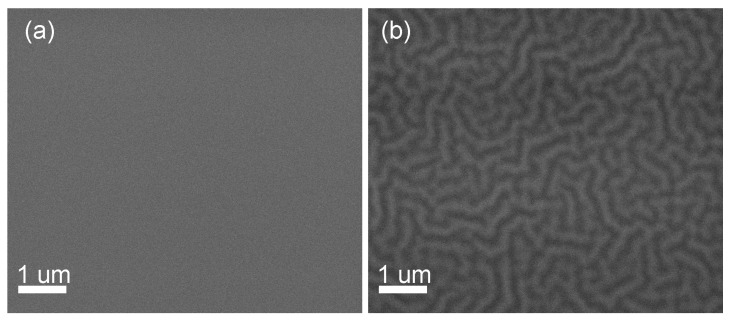
SEM images of the surface of LNO single-layer films after drying at 200 °C, 10 min. (**a**); and 300 °C, 10 min (**b**).

**Figure 4 molecules-28-01938-f004:**
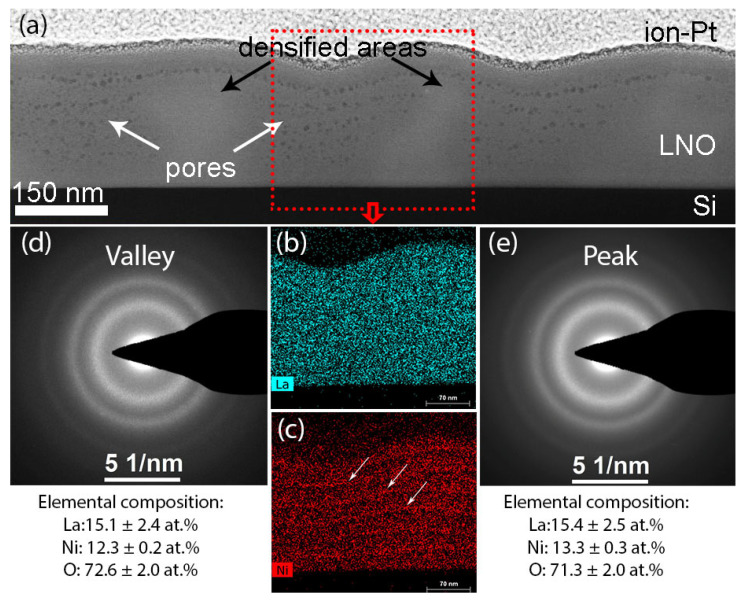
Microstructure of the multilayer LNO film, obtained by layer-by-layer deposition with drying at 300 °C of each layer: HAADF-STEM image (**a**), EDX maps of La (**b**) and Ni (**c**), SAED patterns from the valley (**d**) and from the peak (**e**).

**Figure 5 molecules-28-01938-f005:**
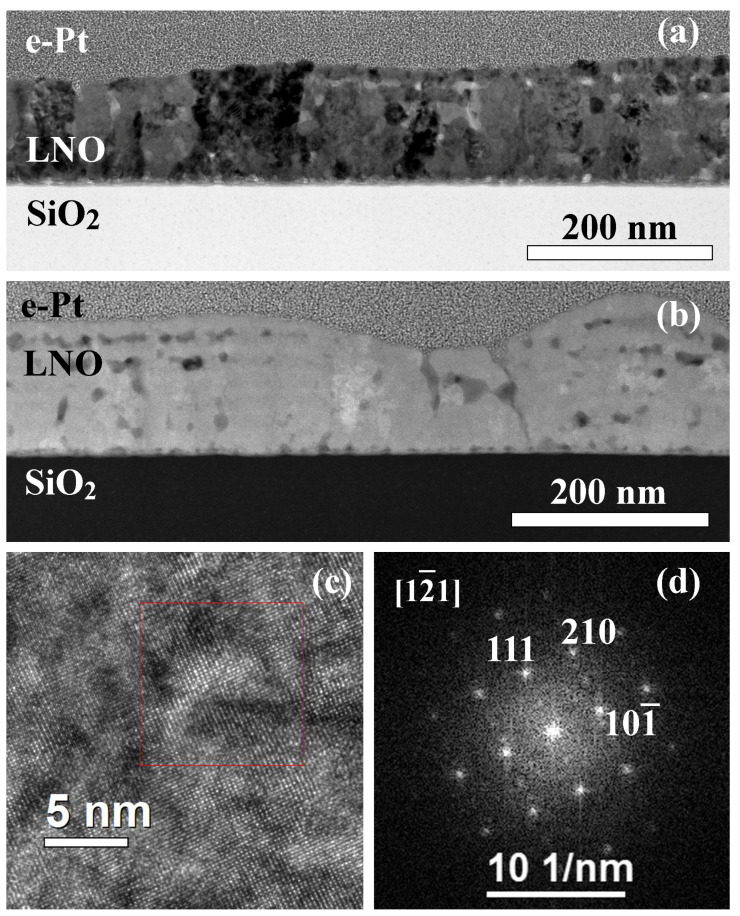
Structure of a LNO-cryst-5 min sample: BF TEM image (**a**), HAADF STEM image (**b**), HRTEM image (**c**), and the corresponding Fourier diffractogram (**d**).

**Figure 6 molecules-28-01938-f006:**
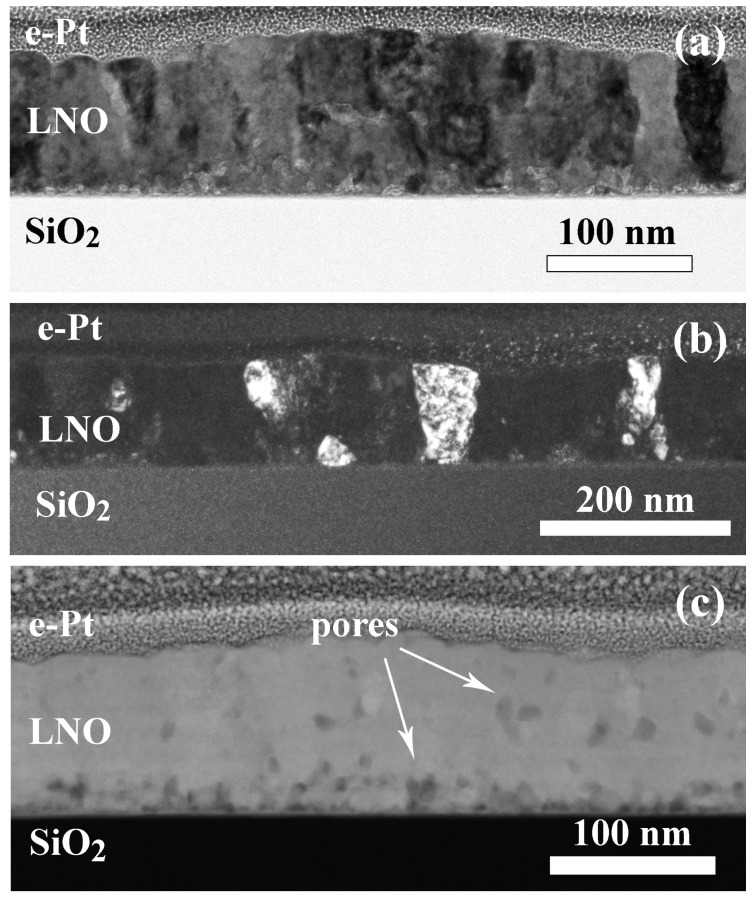
Structure of LNO-cryst-7 min sample: BF TEM image (**a**), DF TEM image (**b**), and HAADF STEM image (**c**).

**Figure 7 molecules-28-01938-f007:**
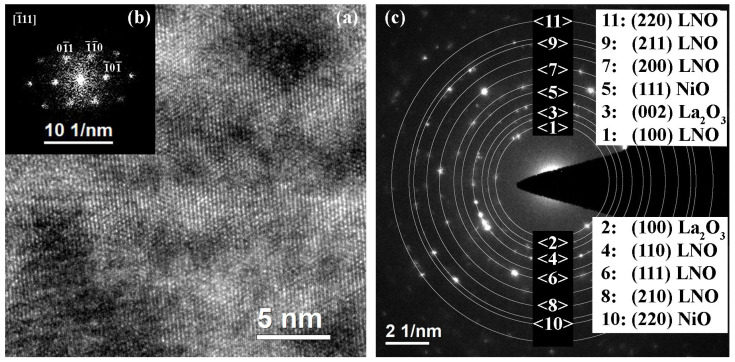
HRTEM image (**a**), corresponding Fourier diffractogram (**b**) and SAED pattern (**c**) of LNO-cryst-7 min sample.

**Figure 8 molecules-28-01938-f008:**
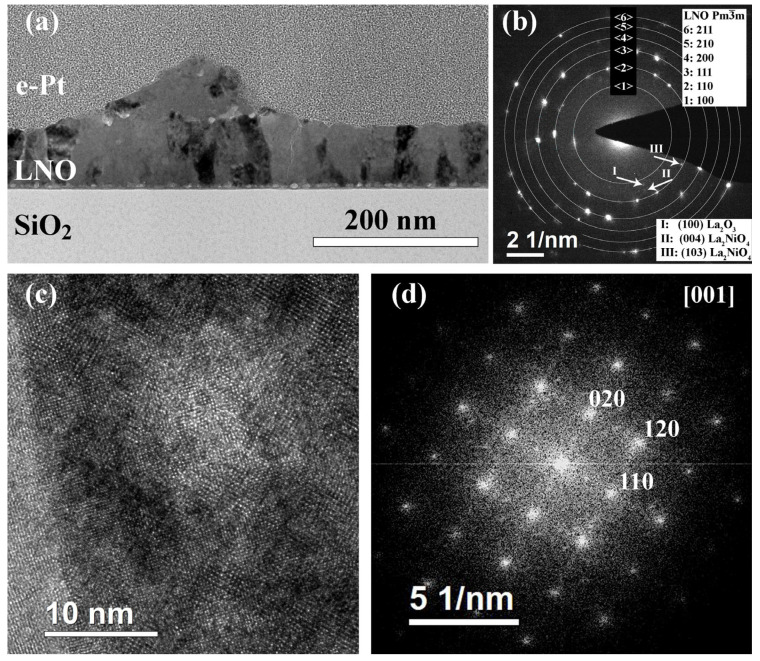
Structure of an LNO-cryst-10 min sample: BF TEM image (**a**), SAED pattern (**b**), HRTEM image (**c**), and the corresponding Fourier diffraction pattern (**d**).

**Figure 9 molecules-28-01938-f009:**
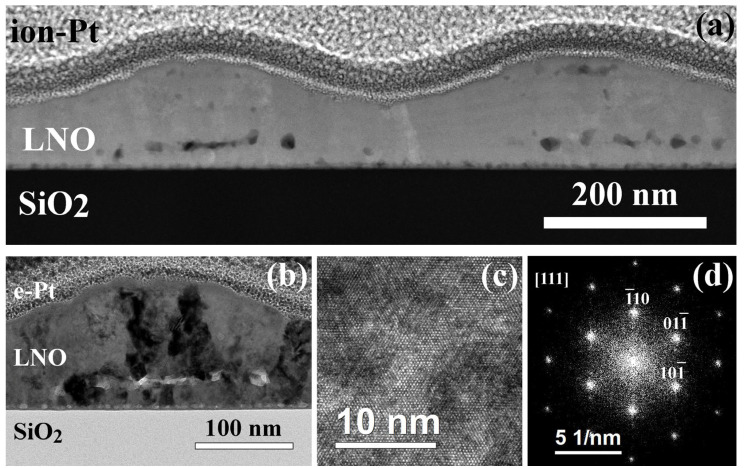
Structure of an LNO-cryst-20 min sample: HAADF STEM image (**a**), BF TEM image (**b**), HRTEM image (**c**), and the corresponding Fourier diffraction pattern (**d**).

**Figure 10 molecules-28-01938-f010:**
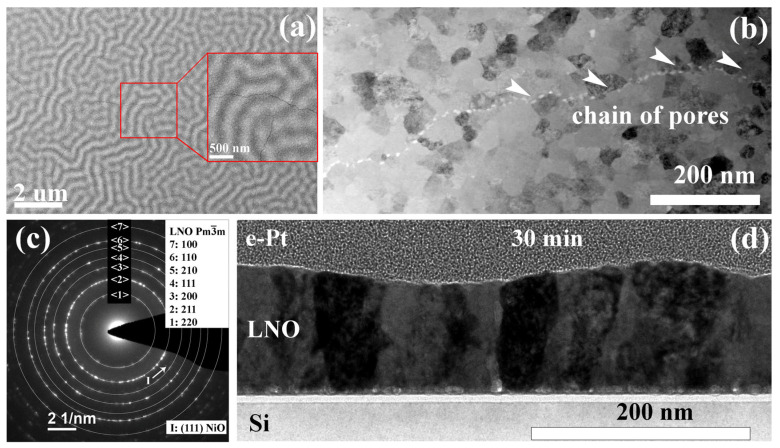
SEM image of the sample surface LNO-anneal-30 min (**a**); plan-view (**b**) and SAED pattern (**c**) of LNO-anneal-90 min, and BF TEM image of a cross section of LNO-anneal-30 min sample (**d**).

**Figure 11 molecules-28-01938-f011:**
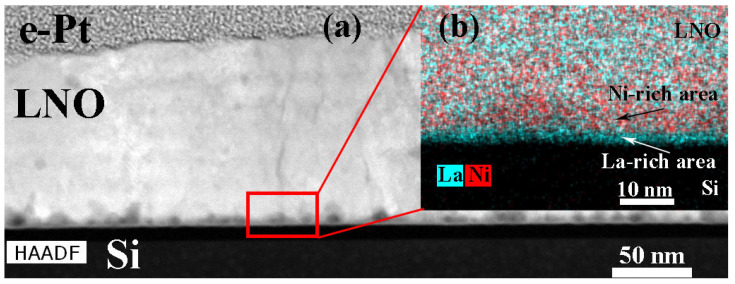
HAADF STEM image (**a**) and elemental map of LNO-anneal-90 min sample (**b**).

**Figure 12 molecules-28-01938-f012:**
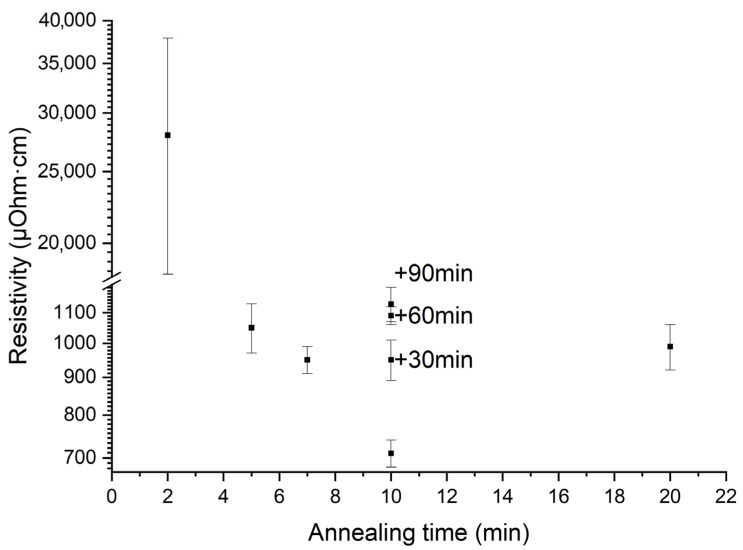
The film resistivity as a function of annealing time of each layer. Samples with additional annealing of the film at 30, 60, and 90 min are shown in at the 10 min annealing time position of each layer during the film preparation. Note: the resistivity scale breaks for a clearer presentation.

**Figure 13 molecules-28-01938-f013:**
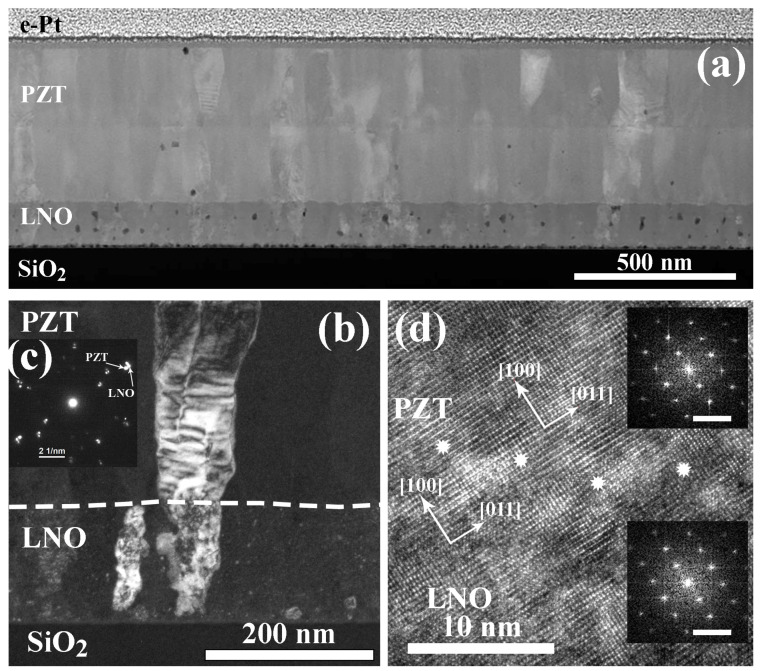
Structure of PZT-LNO-SiO_2_-Si composition: HAADF STEM image (**a**), DF TEM image (**b**), SAED pattern from mutually oriented PZT and LNO grains (**c**), HREM image of PZT-LNO boundary with insertions of Fourier diffractograms (**d**). The mark in the Fourier diffractograms is equal to 5 nm^−1^.

**Figure 14 molecules-28-01938-f014:**
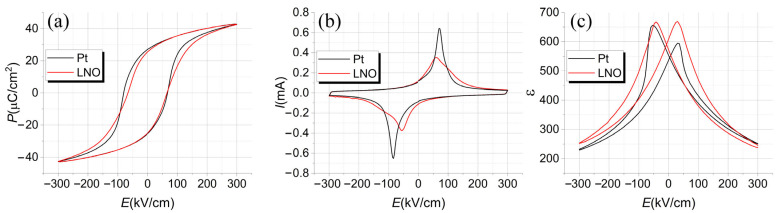
Dependences of polarization *P* (**a**), switching current *I* (**b**) and permittivity *ɛ* (**c**) on electric field strength *E* for PZT films on Pt (black lines) and LNO bottom electrode (red lines).

**Figure 15 molecules-28-01938-f015:**
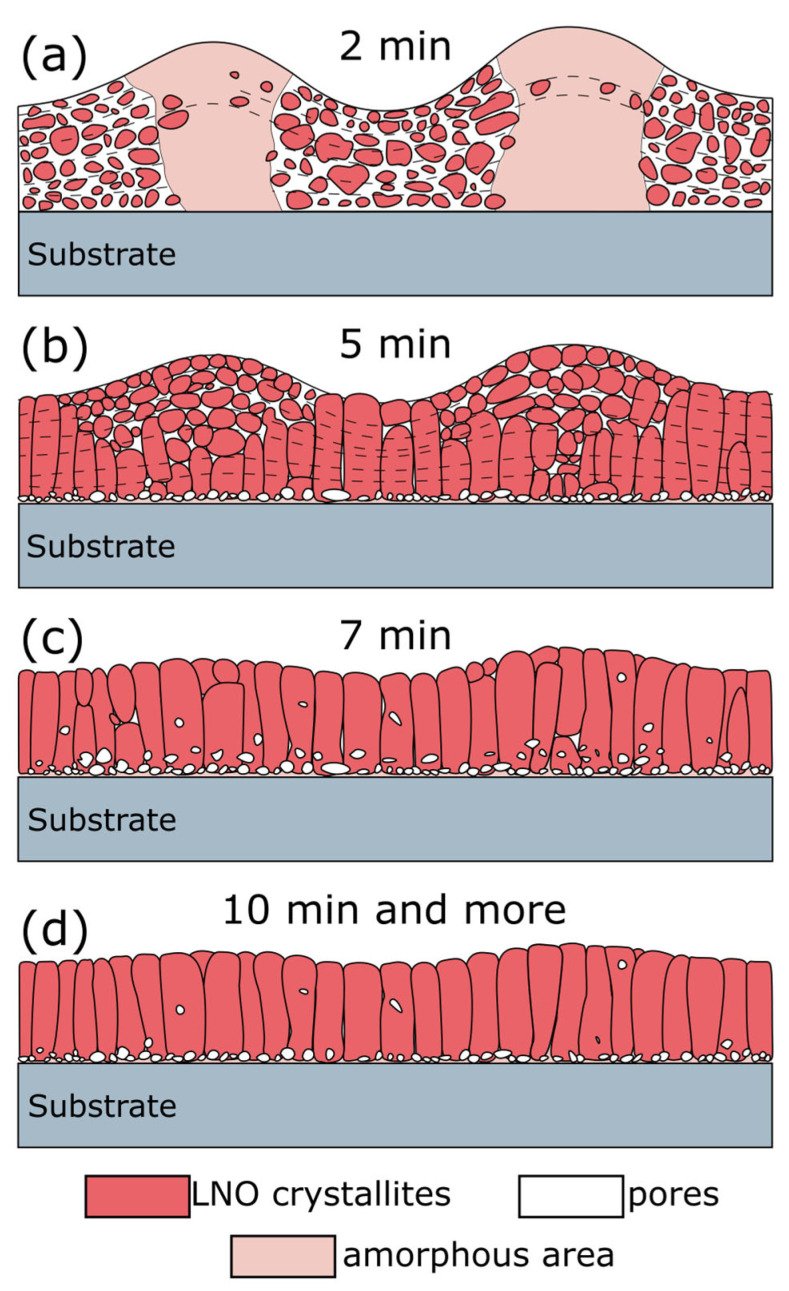
General view of LNO films, prepared at different annealing time of each deposited layer: 2 min (**a**), 5 min (**b**), 7 min (**c**), 10 min, and more (**d**).

**Figure 16 molecules-28-01938-f016:**
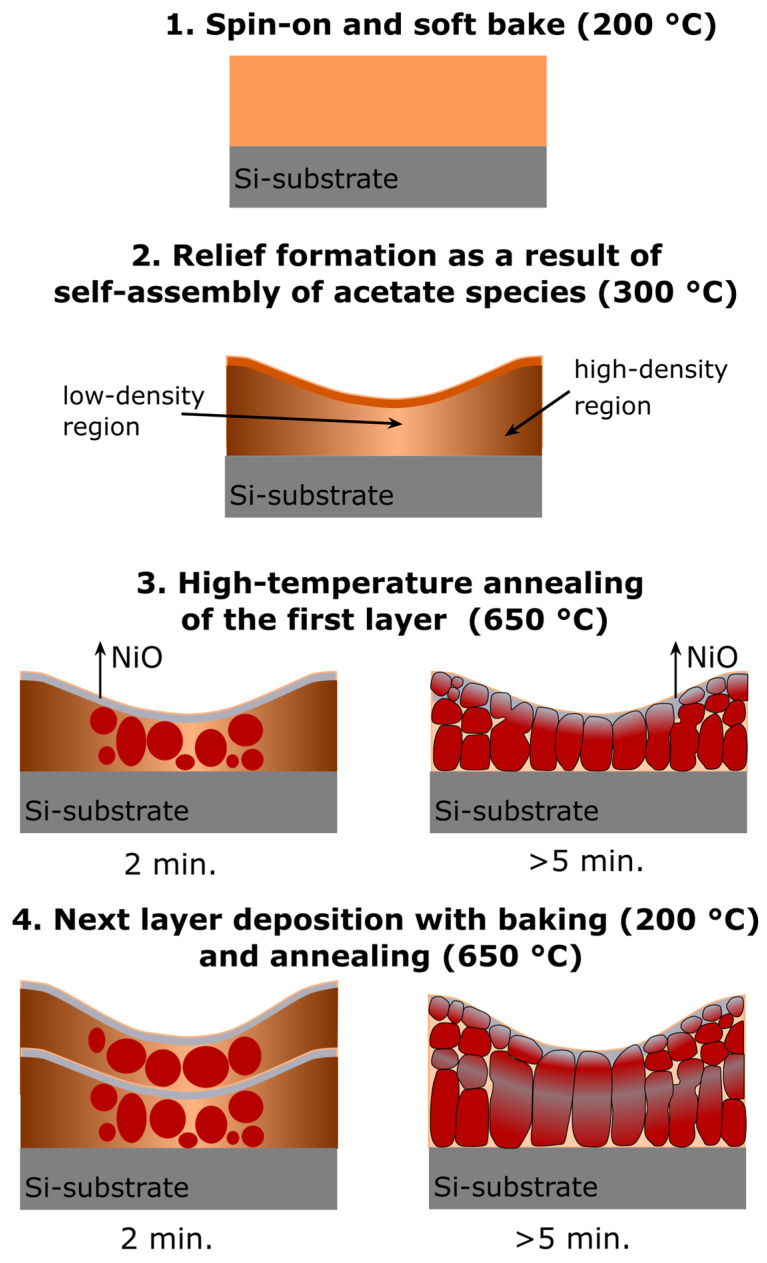
Model of LNO film crystallization.

**Figure 17 molecules-28-01938-f017:**
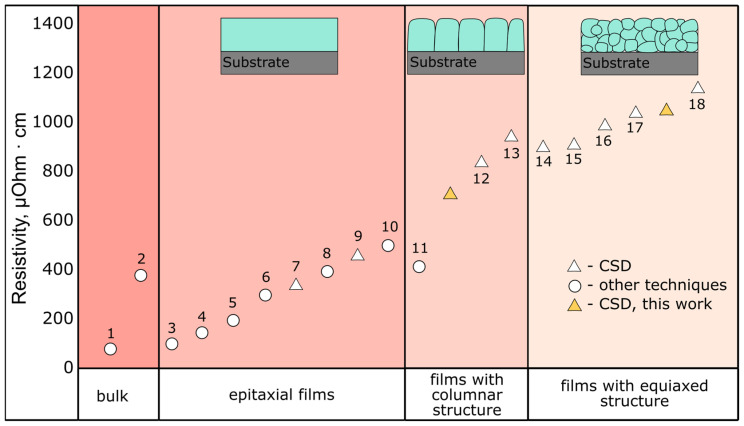
Electrical resistivity of LNO with various internal structures. Values of the resistivity marked with numbers 1–18 are taken from the following references: 1—[88], 2—[89], 3—[28], 4—[92], 5—[30], 6—[20], 7—[90], 8—[30], 9—[90], 10—[29], 11—[27], 12—[22], 13—[15], 14—[42], 15—[19], 16—[41], 17—[22], 18—[19].

**Table 1 molecules-28-01938-t001:** Microstructure and sheet resistance of LNO films prepared at various annealing conditions.

Sample	Crystallization Time, min	Additional Annealing, min	Grain Structure	Main Phase	Resistivity, μOhm·cm
LNO-cryst-2 min	2	no	Equiaxed grains of 6–22 nm and amorphous hillocks. Pores are located between equiaxed grains.	$\mathrm{Amorphous}+{\mathrm{LaNiO}}_{3}\;(Pm\bar{3}m$)	18,000–30,000
LNO-cryst-5 min	5	no	Equiaxed grains of 7–19 nm, columnar grains of 25–63 nm in width, germinated partially or over the entire width of the film. The pores are located mainly between equiaxed grains, as well as at the LNO-SiO2 boundary.	${\mathrm{LaNiO}}_{3}\;(Pm\bar{3}m$)	970–1130
LNO-cryst-7 min	7	no	Predominantly columnar grains of 28–48 nm in width, as well as a small amount of equiaxed grains of 8–18 nm. Pores in the bulk of the film and at the LNO-SiO2 interface.	${\mathrm{LaNiO}}_{3}\;(Pm\bar{3}m$)	900–990
LNO-cryst-10 min	10	no	Columnar grains of 22–38 nm in width in all areas, except of hillocks. Agglomeration of pores locally destroys columnar structure (top part of the film).	${\mathrm{LaNiO}}_{3}\;(Pm\bar{3}m$)	680–730
LNO-cryst-20 min	20	no	Columnar grains of 21–43 nm in width. Agglomeration of pores locally destroys columnar structure (bottom part of the film).	${\mathrm{LaNiO}}_{3}\;(Pm\bar{3}m$)	920–1100
LNO-anneal-(30–90) min	10	30–90	Close to ideal columnar grains of 18–72 nm in size. Inclusions of some single pores and chain of pores.	${\mathrm{LaNiO}}_{3}\;(Pm\bar{3}m$)	890–1010, 1060–1120, 1070–1190 at 30, 60, and 90 min. annealing, respectively

**Table 2 molecules-28-01938-t002:** Electrical data of PZT-Pt and PZT-LNO structures.

Structure	*P*(*E*)				*ε*(*E*)			
	*E*^(−)^, kV/cm	*Pr*^(−)^, µC/cm^2^	*E*^(+)^, kV/cm	*Pr*^(+)^, µC/cm^2^	*E*^(−)^, kV/cm	*ε _max_* ^(−)^	*E*^(+)^, kV/cm	*ε _max_* ^(+)^
PZT-Pt	−84.4	−25.2	68.4	27.2	−54	656	30	595
PZT-LNO	−65	−25	65	25.4	−42	668	30	669

**Table 3 molecules-28-01938-t003:** Samples preparation description.

Experimental Group	Sample Designation	Preparation Description
LNO-SiO_2_(500 nm)-Si Film with variation of annealing time.	LNO-cryst-2 min	Each of the seven LNO layers was soft-baked at 200 °C, 4 min and annealed at T = 650 °C for 2, 5, 7, 10, or 20 min.
	LNO-cryst-5 min	
	LNO-cryst-7 min	
	LNO-cryst-10 min	
	LNO-cryst-20 min	
LNO-SiO_2_(10 nm)-Si Film with additional annealing after deposition of all layers.	LNO-anneal-30 min	Each of the seven layers of LNO was soft-baked at 200 °C, 4 min and annealed at 650 °C, 10 min. Then, the sample was divided into three parts and each one was additionally annealed at 650 °C for 30, 60, and 90 min, respectively.
	LNO-anneal-60 min	
	LNO-anneal-90 min	
PZT-LNO-SiO_2_(500 nm)-Si Ferroelectric film with bottom conducting LNO electrode.	PZT-LNO-SiO_2_-Si	Each of the seven layers of LNO was soft-baked at 200 °C, 4 min and annealed at 650 °C, 10 min, except of the last layer annealed for 30 min. After that, 10 layers of PZT film were deposited. Each layer was dried at 200 °C, 4 min and at 400 °C, 10 min. After deposition of every fifth layer, the film was annealed at 650 °C, 10 min.

## Data Availability

Data are contained within the article. The raw data presented in this study are available on request from the authors.

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
