# Peer review of "Control of Columnar Grain Microstructure in CSD LaNiO3 Films"

_molecules, 2023, doi:10.3390/molecules28041938_

Round 1
Reviewer 1 Report
The authors systematically studied the mechanism of columnar grain growth of LNO perovskite films on Si/SiO2 wafer substrates via chemical solution deposition methods. In previous studies, columnar grain growth of perovskite-based systems on amorphous substrates has been investigated in terms of the dependence of sample structure/properties on substrate temperature and precursor solution concentrations (10.1021/acs.cgd.1c00331). The authors focused on vertical heterostructures consisting of similar films, which should offer hints on future ferroelectronics and related physics. The authors conducted a thorough set of experiments to understand the microstructures of their samples, but before publication, they still need to address the following comments/suggestions.
1. The length of the introduction section is unnecessarily too long. The authors explained much background information in unexpected details, but a great portion of the explanations are generally commonly known and are supposed to appear in related textbooks. This actually happens also in other parts, for example, considering the sentence in line 588: “…film structure has strong effect on the LNO resistivity.” This is just another kind of expression of the generally admitted structure-property relation. Therefore, it is suggested that the introduction and other parts be more organized and briefly summarized, and more attention be drawn on topics that are challenging and not fully understood/achieved in the field. The authors are also suggested to emphasize more on the significance of their studies on applications, as well as avoiding spoken-English-style expressions.
2. Solution-assisted crystal growth methods are highly promising in achieving improved sample qualities and novel structures, and are indeed attracting attentions among researchers. Solution-assisted methods are also being applied to conventional synthesis such as CVD (e.g., 10.1021/acsnano.9b09857). Therefore, the authors are encouraged to give more discussions on their selection of the CSD growth technique.
3. Did the authors introduce any kind of protection gases to the growth? Usually this is necessary for high quality synthesis.
4. In order to understand the formation of the wavy surface morphology in their samples, a comparison between such kind of structure and other possible ones (e.g., purely flat surfaces) in terms of formation energy can be conducted by means such as DFT calculations. Do the authors have similar future research plans?
5. In line 260, should it be “As a result” instead of “As a reason”?
6. From the electron diffraction patterns, it seems that the (210) ring is missing in LNO-cryst-2min samples, but appears in all the other ones. Why?
7. In line 395, why will longer annealing process lead to increased mechanical stress? Generally thermal treatment is a technique to release the as-grown strains in materials.
8. Figure 15 is simply a reprint of parts in Figures 1, 5, 6, and 9, which is improper and is suggested to remove.
Reviewer 2 Report
The article describes the microstructural evolution and sheet resistance of LaNiO3 films as a function of processing condition. The works seems reasonably well done and should be of interest. I have a few minor corrections and comments.
Figure 5d – the zone axis of the FFT was not indicated
Figure 9d – Verify the zone axis label and labeled reflections are correct, currently the zone law is violated
Line 352 “90min” Add a space between 90 and min
On lines 212 and 213 it’s stated that, “These fluctuations are likely formed due to NiO evaporation”. However, on lines 359 and 360 it’s written, “The La:Ni ratio according to elemental mapping is approximately 1:1”. Are these statements mutually exclusive? If not, please provide some greater explanation. Providing additional details on how the EDX data was quantified would be helpful too.
When dimensions are reported a mean value is provided. An uncertainty value is also included. Please describe what that value indicates, is it one standard deviation about the mean? Also please include the number of data points measured to arrive at this mean value.
Is there any preferential crystal growth direction of the columnar grains (i.e., texturing) and how does it evolve as a function of processing condition?
On line 250 to 252, “crystallized mainly with the formation of the cubic perovskite phase LaNiO3 (s.g. ??3Ì…?), although it should be noted that part of the Fourier diffractograms can be solved using both the cubic and trigonal LaNiO3 (?3Ì…?).”. It’s my understanding, which my not be correct, that LaNiO3 forms as ?3Ì…?, but it can be approximated as a psuedocubic (??3Ì…?) unit cell. Therefore, this description seems confusing as it’s making it sound as if LaNiO3 can crystal into two distinct structures. I would recommend clarifying this point and ensuring the text conveys a clear and accurate message.
Reread the manuscript for grammar. There are numerous errors.
I don’t think the following articles were included and may be worth considering:
DOI: 10.1111/j.1151-2916.2003.tb03555.x
DOI: 10.1016/S0955-2219(03)00382-0
Reviewer 3 Report
The paper presents interesting results of structure and Electrical properties in LNO films deposited by CSD techniques,the following questions need to be revised before publication:
1 In Table 1, why two kinds of SiO2 thickness of 10nm and 500nm are used?
2 Why are cracks studied in Figure 10? It seems that cracks are not related to the whole article. Are there no cracks in other samples?
3Does the surface hillocks relief affect the measurement of film resistivity with different parameters?
4 In Table 2, resistivity should also be used instead of resistance in accordance with Figures 12 and 17.
